# Functional reorganisation of the cranial skeleton during the cynodont–mammaliaform transition

Stephan Lautenschlager [1✉], Michael J. Fagan [2], Zhe-Xi Luo [3], Charlotte M. Bird[1], Pamela Gill[4,5] & Emily J. Rayfield [5✉]

Skeletal simplification occurred in multiple vertebrate clades over the last 500 million years, including the evolution from premammalian cynodonts to mammals. This transition is characterised by the loss and reduction of cranial bones, the emergence of a novel jaw joint, and the rearrangement of the jaw musculature. These modifications have long been hypothesised to increase skull strength and efficiency during feeding. Here, we combine digital reconstruction and biomechanical modelling to show that there is no evidence for an increase in cranial strength and biomechanical performance. Our analyses demonstrate the selective functional reorganisation of the cranial skeleton, leading to reduced stresses in the braincase and the skull roof but increased stresses in the zygomatic region through this transition. This cranial functional reorganisation, reduction in mechanical advantage, and overall miniaturisation in body size are linked with a dietary specialisation to insectivory, permitting the subsequent morphological and ecological diversification of the mammalian lineage.

[1] School of Geography, Earth and Environmental Sciences, University of Birmingham, Birmingham, UK. [2] Department of Engineering, University of Hull, Hull, UK. [3] Department of Organismal Biology and Anatomy, University of Chicago, Chicago, USA. [4] Earth Sciences Department, The Natural History Museum, London, UK. [5] Bristol Palaeobiology Group, School of Earth Sciences, University of Bristol, Bristol, UK. ✉email: s.lautenschlager@bham.ac.uk; e.rayfield@bristol.ac.uk

Throughout its 500-million-year history, the evolution of the vertebrate skull has been characterised by recurring episodes of simplification[1–3]. Assumed to increase cranial integrity and skull strength[4], simplification has been achieved through solidification, fusion, reduction, and loss of skeletal elements. These simplification events can be found across different tetrapod lineages[5–7] and throughout synapsid history, including mammals and their synapsid, pre-mammaliaform cynodont precursors (referred to as cynodonts hereafter)[8]. The evolution of mammals from cynodonts is a key transition in vertebrate history and is further characterised by a number of modifications of the cranial structure[9], including the evolution of a novel, secondary jaw joint and a reduction of the seven bones in the lower jaw to a single tooth-bearing bone in crown mammals[10,11]. Furthermore, this modification of the hard-tissue structures is intimately linked to the rearrangement of the jaw musculature[12], which is assumed to have led to the evolution of a feeding system with a more effective transfer of muscle forces to bite forces[13]. Parallel to the reorganisation of the mandible, further modifications took place in the skull, such as the loss of the postorbital bar, the development of a secondary palate, and the integration of the quadrate into the middle ear[9,14–16]. These modifications are thought to have enhanced the structural integrity and strength of the cranium, supposedly in response to the increased forces generated by a more powerful and complex jaw musculature and the ecological need for stronger bite forces[9,10,12,17]. Consequently, it has been argued that the selection pressure for stronger skulls could represent one possible driver for the morphological transformation across the cynodont–mammaliaform transition[9] (Fig. 1).

However, this proposition has not been tested quantitatively. It is further unclear how, on the one hand, there existed a trend towards strengthening of the skull, while, on the other, reductions of the bones supporting the musculature could potentially weaken cranial strength. Furthermore, a recent study has demonstrated that reduction in overall skull size, rather than shape changes of the cynodont and mammaliaform mandibular complex, was one of the key mechanisms for functional optimisations[18]. Here, we test the hypothesis that the cranial skeleton did indeed become stronger across the cynodont–mammaliaform transition, using a combination of digital reconstruction methods and biomechanical analysis techniques. This allows us to determine whether skulls experience progressively lower magnitudes of cranial stress while increasing their bite forces during feeding as the skulls become more 'mammal-like.' We further analyse the biomechanical performance of different anatomical regions to test for possible selective functional optimisation of different modifications of the cranial skeleton.

## Results
The biomechanical analyses indicate that there is no overall trend towards decreasing cranial stress and strain magnitudes across the cranial skeleton of different cynodont and mammaliaform taxa as a whole. The FE contour plots show a generally similar pattern of stress distributions for all taxa (Figs. 2 and 3), with the exception of *Hadrocodium wui*, which experiences very high cranial stresses in all bite scenarios (Figs. 2k, l and 3k, l). However, some nuanced functional patterns can be distinguished between the studied taxa. First, a concentration of (mostly compressive) stresses on the skull roof and the orbital region close to the bite point in the cynodonts *Thrinaxodon liorhinus*, *Chiniquodon sanjuanensis*, cf. *Probainognathus*, and, to a lesser extent, *Diademodon tetragonus* (Figs. 2a–h and 3a–h). Second, in the mammaliaforms *Morganucodon oehleri* and *Hadrocodium wui*, as well as in the extant taxa, the stress hotspots shift towards the zygoma (Figs. 2i–l and 3i–l). However, it should be noted that stresses are substantially

increased overall in *H. wui* compared to all other taxa. This pattern is also observed when the skull is considered as a single unit. Stress magnitude distributions are very similar across the different taxa, with *Morganucodon* exhibiting mostly a large number of elements with low stresses, whereas *Hadrocodium* has an increased number of elements with moderate to high stresses for all tested scenarios (Supplementary Fig. 1).

Biomechanical results obtained for the individual cranial regions (Fig. 4) demonstrate a similar pattern to those more qualitatively observed in the contour plots. The narial and the frontal regions (Fig. 4a, b) show a largely constant distribution across the studied taxa and bite scenarios, whereas values for the skull roof and the braincase (Fig. 4c, e) are higher in the cynodonts compared to the *M. oehleri* and the extant taxa. An opposite trend can be observed for the zygoma (Fig. 4d), with *T. liorhinus*, *D. tetragonus*, and *C. sanjuanensis* showing consistently lower stress values than the more derived taxa. Again, *H. wui* presents an outlier with high stress magnitudes throughout. The quantification of deformation experienced by the individual cranial regions (Fig. 5) for the different bite scenarios confirms the above results, in particular for the zygoma. Deformation increases from the cynodont to the mammaliform taxa (Fig. 5g, j) before reaching low magnitudes in the extant taxa. Additional tests were performed for *H. wui*, due to the high stress magnitudes experienced in the zygomatic region and the fact that the zygoma had to be reconstructed to a large extent for this specimen. However, regardless of the morphology of the reconstructed zygomatic region recorded stress values were more than 100% higher than in the other species (Supplementary Fig. 2).

A comparison of bite forces across the studied taxa shows that absolute bite forces decrease from the cynodonts towards the mammaliaforms and increase again in the extant taxa (Fig. 6a). This is unsurprising considering the considerable size differences between the taxa. However, the mechanical advantage (=bite forces relative to muscle force) shows a very similar, albeit somewhat less pronounced pattern (Fig. 6b). Quantified against dietary regimes, insectivorous taxa have both the lowest absolute bite forces (Fig. 6c) and mechanical advantage (Fig. 6d). Herbivorous taxa, conversely, have the highest absolute and relative bite forces, but it should be noted that our sample includes only one fossil and one modern herbivorous species.

## Discussion
As illustrated by the results of the finite element analyses (FEAs), there is no general trend towards an increase in cranial strength and biomechanical efficiency across the cynodont–mammalian transition. Rather, the functional differences observed here are more nuanced and specific to individual anatomical regions, suggesting that the cranial skeleton underwent a selective reorganisation. Such modular evolution of the vertebrate skeleton has been demonstrated to constitute an important factor in mammals:[19] developmental[20,21], molecular[22,23] and morphological[24–26] studies have found evidence for a semi-autonomous trait evolution of mammalian cranial regions. In particular, interconnected anatomical and functional modularity was found to be present in the evolution of the mammalian middle ear due to a reduction in cranial complexity[26]. Our results indicate that a similar degree of functional modularity was present in the cranial skeleton of different cynodont and mammaliaform taxa in the context of skull strength and resistance to mastication forces. Historical studies hypothesised that the cynodont–mammaliaform evolution favoured an increase in the structural integrity of the crania in response to the emergence of a new and more efficient jaw muscle arrangement, which conveyed an evolutionary advantage[9,17].

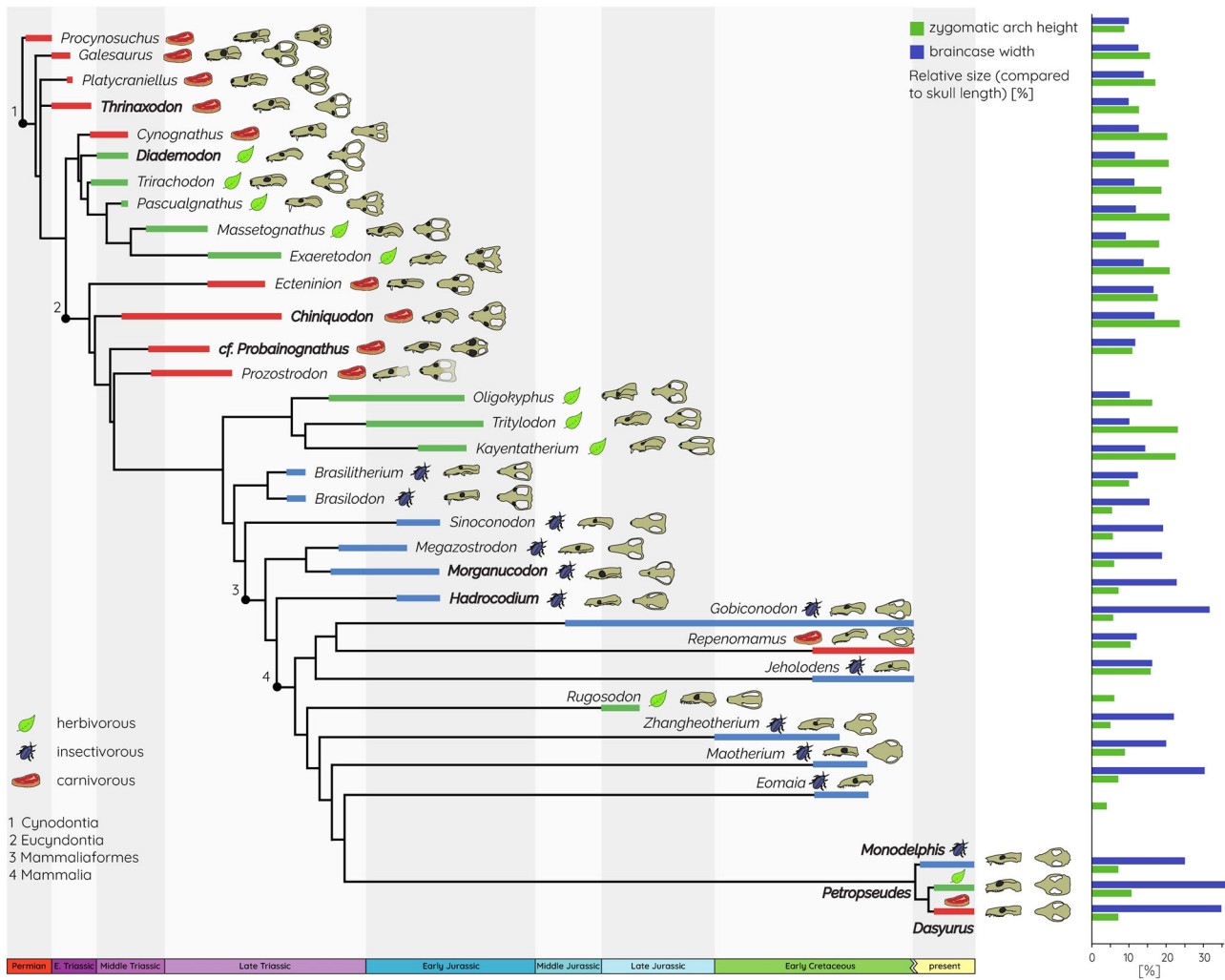

**Fig. 1 Evolutionary relationship of cynodonts and mammaliaforms depicting dietary adaptations and modifications in skull morphology.** Relative changes in the height of the zygomatic arch and the width of the braincase (compared to skull length) are shown in the bar graph (see Supplementary Table 1 for details). The phylogeny is simplified after previously published phylogenies[51,60,61], and dietary categories were assigned according to literature data[52,62–64]. Taxa included in the biomechanical analysis are highlighted in bold.

If skull evolution favoured a general enhancement of skull integrity, it would be predicted that those components/modules associated with muscle attachment (i.e., zygoma, skull roof) would all show decreasing stress and strain magnitudes. However, the biomechanical analyses offer a different picture, in which the zygoma experiences increased stress magnitudes in the mammaliaform taxa, while a reduction of stresses occurs in the skull roof and the braincase.

Although both trends are correlated with a rearrangement of the jaw muscle attachment to the zygomatic arch (i.e., the dominant role of masseter muscle complex)[12,27], this modification does not appear to be accompanied by an osteological reinforcement of the zygoma to counteract increased muscle forces. Instead, the zygomatic arch underwent a considerable reduction in girth and vertical height in mammaliamorphs and mammaliaforms, including *Morganucodon oehleri* and *Hadrocodium wui*[28] (Fig. 1). This presents an obvious conundrum, as the reduction of load-bearing bones and the increase in stress magnitudes are functionally disadvantageous.

At the same time, stresses are deflected from the skull roof and the braincase by the rearrangement of the adductor musculature. Skull roof bones adapt to 'internal' (i.e., growth and development of the brain and sensory organs) and 'external' forces (i.e., muscle pull and bite forces)[29–31]. Consequently, reduced loads (and therefore stresses) on the braincase could have been critical in allowing these regions to become larger with the increase in brain size[15] (Fig. 1). *Morganucodon oehleri* and *H. wui* represent successive stages of brain expansion[32], which might have benefited from reduced loads on the skull roof and the braincase, allowing plasticity of these regions. However, *H. wui* records the highest stress and strain magnitudes, in particular in the zygoma and the skull roof. *Hadrocodium wui* differs from its precursors in an enlarged brain vault, a mediolaterally wide but dorsoventrally strongly flattened skull, but not fully detached mammalian middle ear[15,33–35]. We interpret that a further expansion of the posterior skull region made its skull more susceptible to stresses and strains induced by the jaw musculature (i.e., the temporalis muscles), as shown by our FE analysis.

The rearrangement of the jaw adductor musculature appears not to have resulted in a more efficient transfer of muscle force to bite force, as had previously been postulated[13]. As demonstrated here, absolute and relative bite forces (=mechanical advantage) were found to decrease in the studied mammaliaform taxa relative to those of their cynodont precursors, but also in comparison to the studied extant taxa (Fig. 6a, b). This is consistent with findings for bite force estimates obtained from mandible models

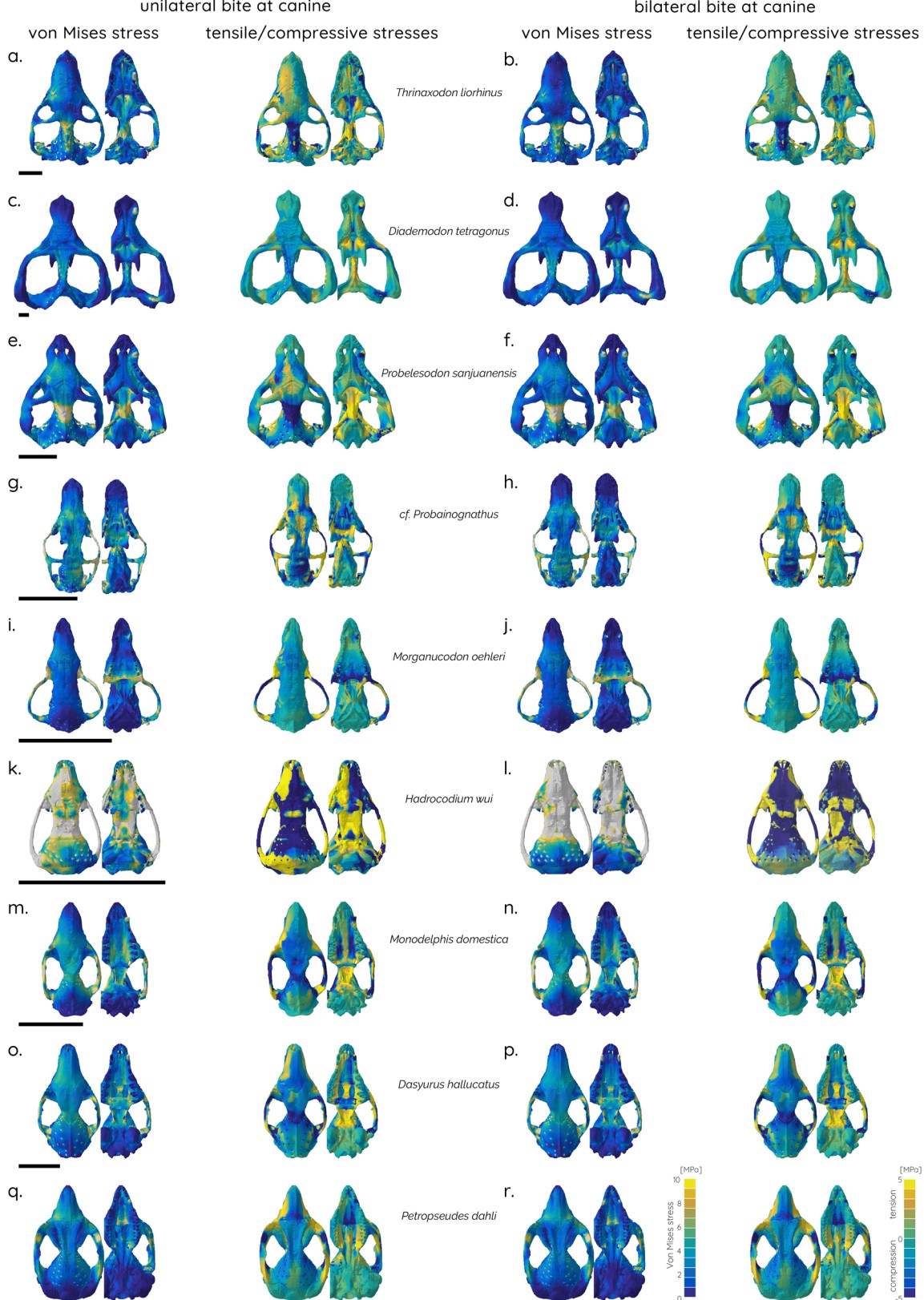

**Fig. 2 Finite element analyses results for studied taxa.** Contour plots of von Mises stress (left) and principal tensile/compressive stresses (right) of skull models in dorsal and ventral. Unilateral bite at left canine tooth (**a**, **c**, **e**, **g**, **i**, **k**, **m**, **o**, **q**) and bilateral bite at canine teeth (**b**, **d**, **f**, **h**, **j**, **l**, **n**, **p**, **r**). Scale bar equals 10 mm.

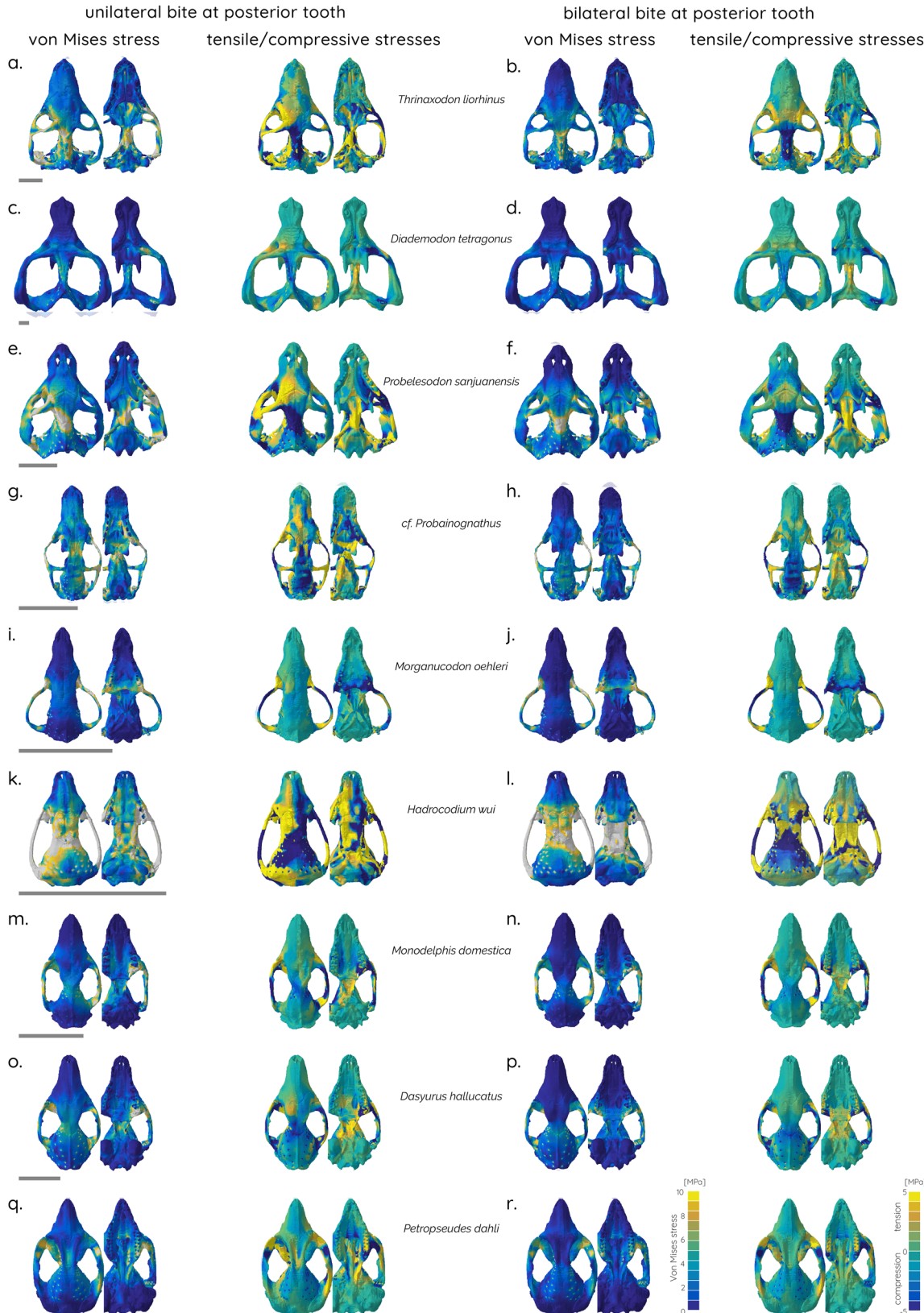

**Fig. 3 Finite element analyses results for studied taxa.** Contour plots of von Mises stress (left) and tensile/compressive stresses (right) of skull models in dorsal and ventral. Unilateral bite at left posterior tooth (**a**, **c**, **e**, **g**, **i**, **k**, **m**, **o**, **q**) and bilateral bite at posterior teeth (**b**, **d**, **f**, **h**, **j**, **l**, **n**, **p**, **r**). Scale bar equals 10 mm.

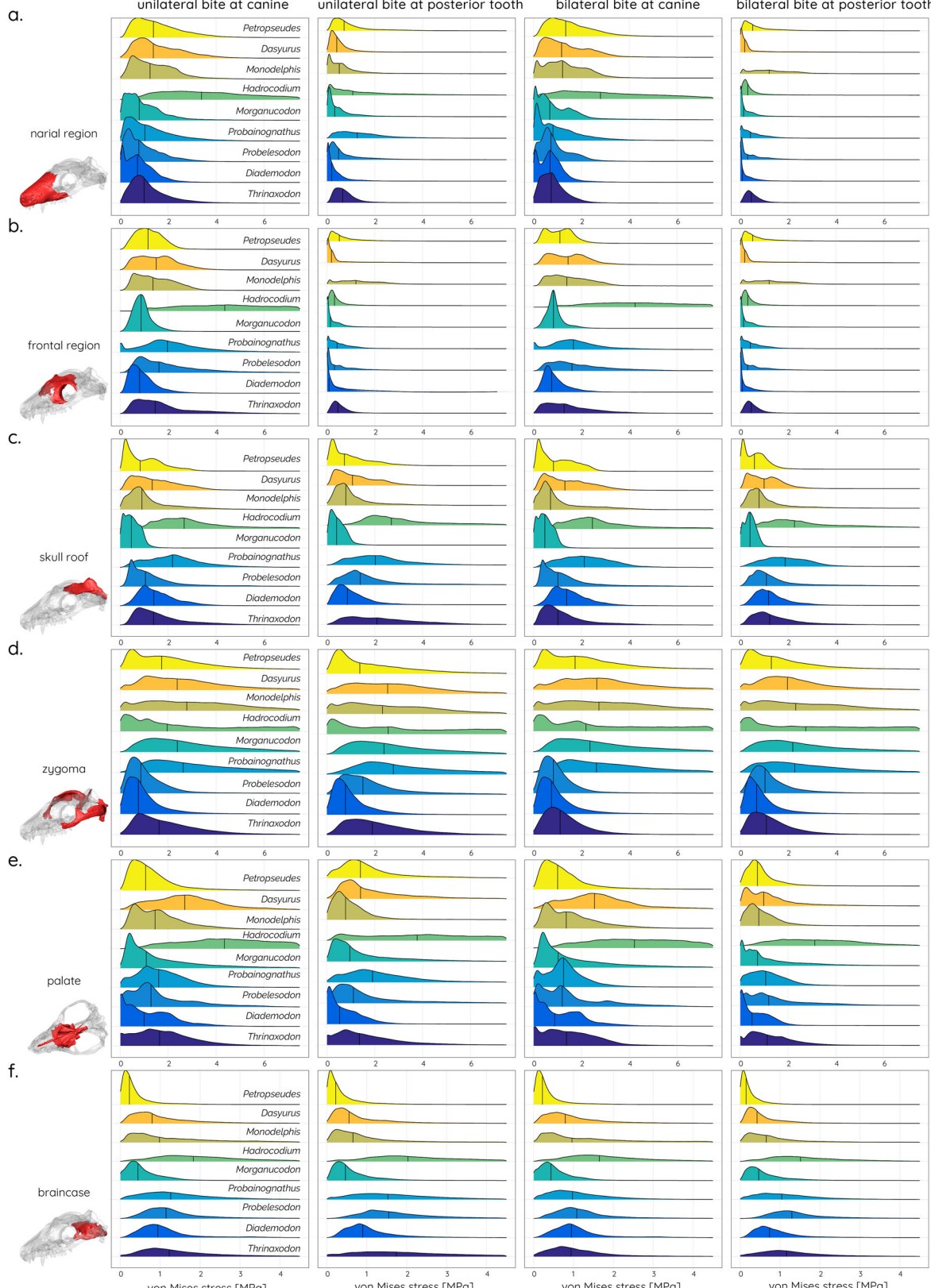

**Fig. 4 Stress distribution across different cranial regions.** Ridgeline plots showing the distribution of von Mises stress for the **a** narial, **b** frontal, **c** skull roof, **d** zygoma, **e** palatal, and **f** braincase regions.

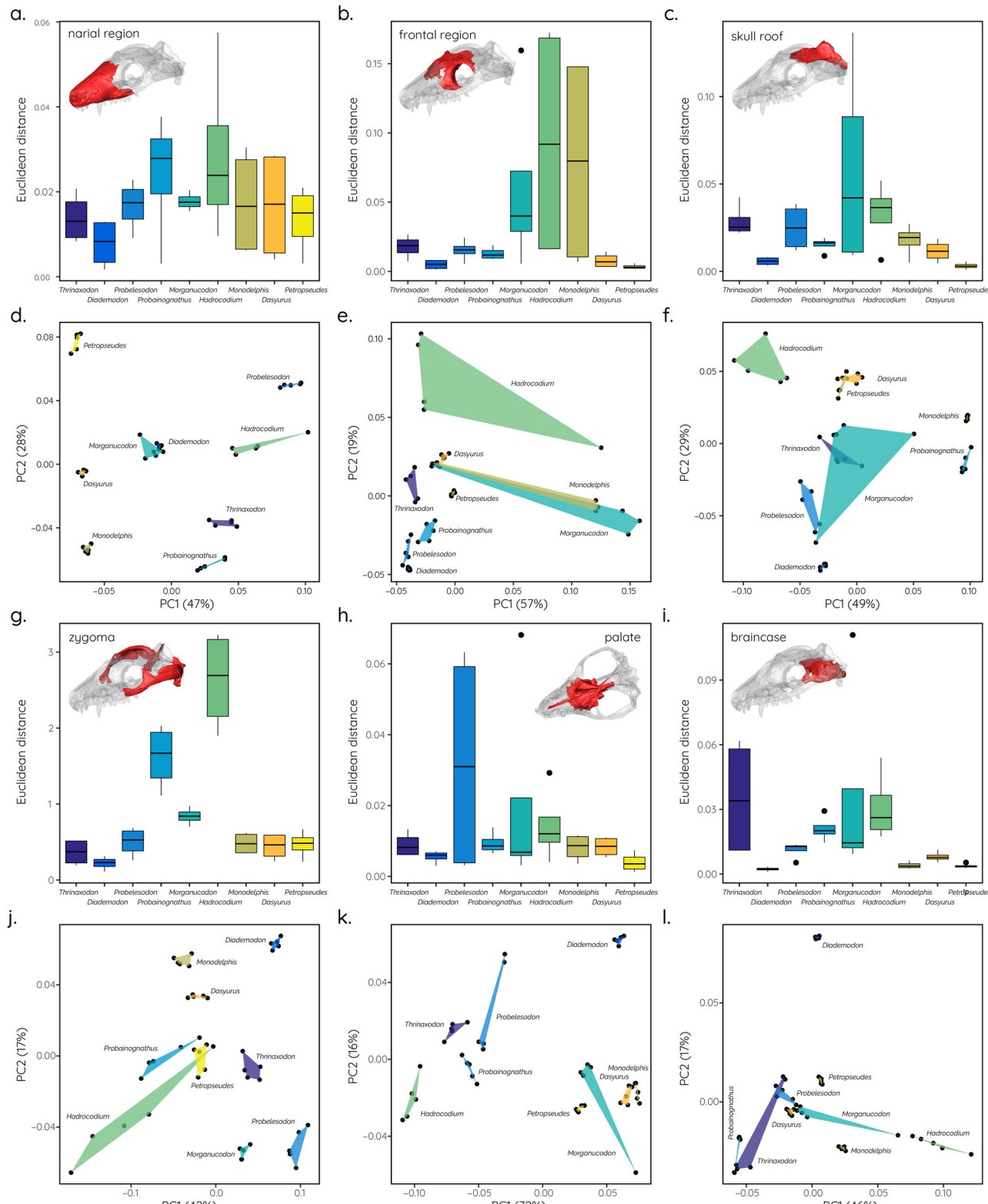

**Fig. 5 Deformation in cranial models quantified using a landmark-based approach.** Boxplots showing deformation of individual cranial regions compared to undeformed models based on the Euclidean distances between PCs 1–3 (**a**, **b**, **c**, **g**, **h**, **i**) and corresponding PCA plots (**d**, **e**, **f**, **j**, **k**, **l**). Results combine all tested bite scenarios (unilateral and bilateral canine bite, unilateral and bilateral posterior bite, undeformed and deformed models, see Supplementary Table 2 for details).

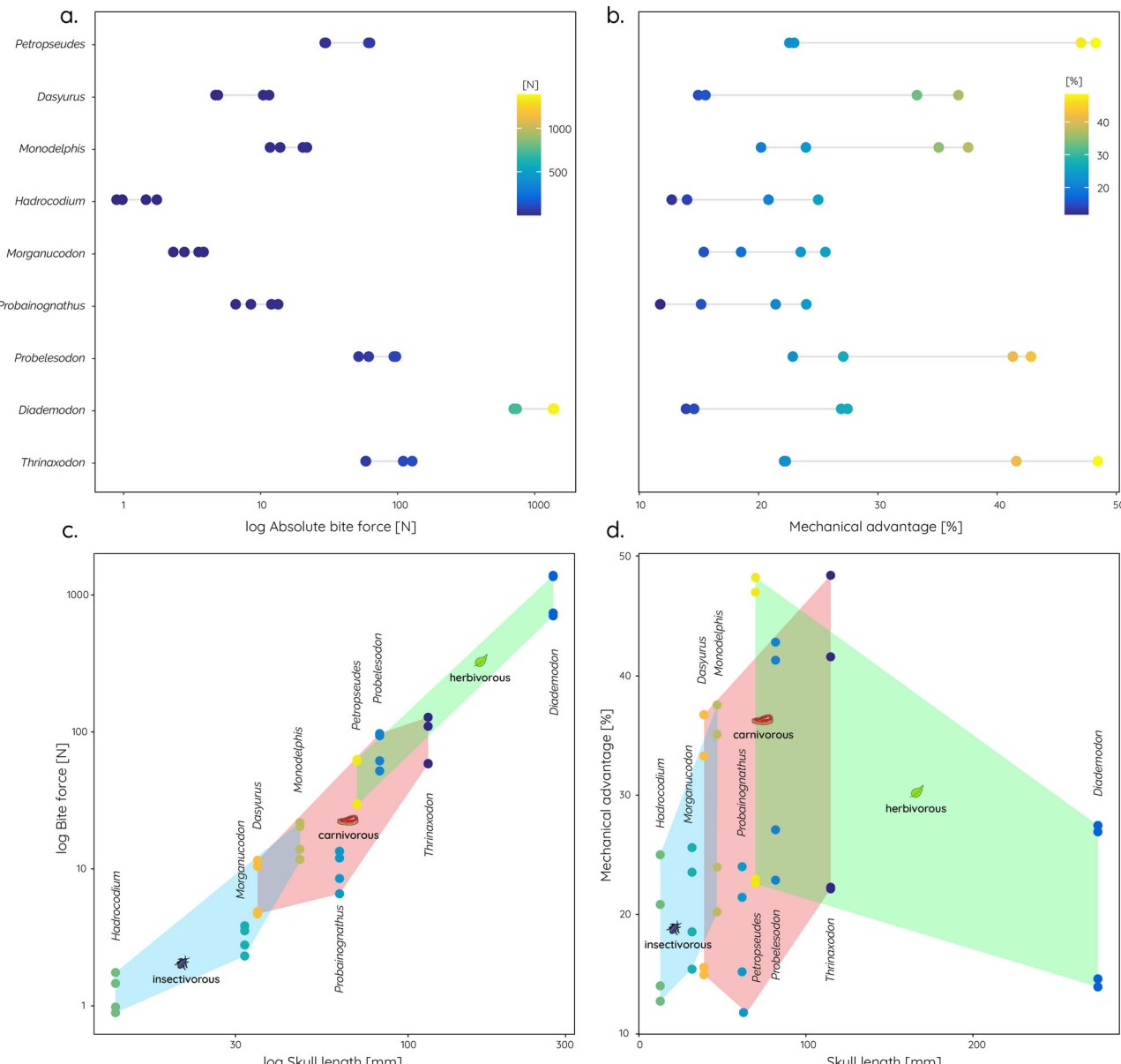

**Fig. 6 Bite forces and mechanical advantage of studied taxa.** Absolute bite forces in newtons N (**a**) and mechanical advantage, bite force relative to input muscle force (**b**), obtained from the finite element analyses for all taxa and bite scenarios. Correlation between bite force and skull length (**c**) and between mechanical advantage and skull length (**d**) with convex hulls indicating dietary adaptations.

of the same taxa[18]. However, the shift of parts of the jaw musculature to the zygomatic arch may have had a different advantage. The reorientation of the force vector in an anterior and vertical direction associated with the differentiation of the neomorphic masseter muscles was hypothesised to help reduce joint reaction forces in some studies using theoretical models[36,37]. Recent quantitative modelling does not support this assumption, though[12,18]. Despite the negative impact of the muscle reorientation on the loading of the zygomatic arch, as shown here, the functional benefits of shifting stress from the skull roof and the braincase could have outweighed these disadvantages. Many modern mammals possess temporal fasciae attaching dorsally to the zygomatic arch and bracing it against the ventrally directed muscle loading of the masseter with a minimum of material[38,39]. The reorientation of the jaw musculature (in particular the masseter complex) could have had another advantage unrelated to maximising bite force. Greater mobility of the mandible[10,13]

and precise tooth occlusion[40] may have required precise muscle control to allow more complex jaw movements, including long-axis rotation (perpendicular to the jaw hinge axis). Such rolling of the jaw was recently identified as an essential mechanism for tribosphenic molar function[41].

Due to the quality and resolution of the CT datasets, cranial sutures could not be identified consistently across the studied taxa and were therefore omitted from the FEA models. However, it cannot be fully ruled out that sutures between bony elements might have further reduced stresses in the skull in concert with other soft-tissue structures. We further note that the specimen of cf. *Probainognathus* is very likely a juvenile; in our analysis, its gracile zygomatic arch may lead, therefore, to a potential underestimation of the stresses on this structure.

In this context, it is noteworthy that diet appears to have a substantial influence on the skull and, in particular, on zygomatic arch shape and size, as well as on absolute and relative bite forces.

Most cynodonts, whether carnivorous or herbivorous, possessed dorsoventrally high and expanded zygomatic arches, with relative sizes of 20–30% of the skull length (Fig. 1). In contrast, taxa close to the cynodont–mammaliaform transition tend to show more reduced zygomatic arches (5–10% of skull length) and are assumed to have been adapted to a predominantly insectivorous diet[42,43]. As shown by the results, insectivorous taxa also have the smallest skulls (Fig. 6c, d). This is consistent with evolutionary patterns in modern marsupials, in which insectivorous taxa sit at the lower end of the size range, while carnivorous and omnivorous taxa are predominantly larger in comparison[44]. Similarly, mandibular morphology has been shown to respond selectively in terms of regional modification with diet in modern mammals[45,46]. This is consistent with the functional differences in the cranial regions in the extant taxa with different dietary regimes in our sample (Figs. 4 and 5).

The change in zygomatic arch height parallels ontogenetic niche shifts observed in gomphodontian cynodonts. For example, smaller individuals (inferred to be juveniles) of *Exaeretodon argentinus* and *Exaeretodon riograndensis* had a low zygoma adapted to crushing-dominated feeding, whereas larger individuals (inferred to be adults/older ontogenetic stages) showed a high zygoma adapted to a chewing-dominated feeding style[47]. A similar trend has been found tentatively also in the more basal cynodonts *Galesaurus* and *Thrinaxodon*[48,49]. A specialisation of the musculoskeletal system towards a crushing function is commonly correlated with an adaptation to faunivory (including insectivory). The skeletal miniaturisation, reorganisation of individual skull regions (including the zygoma), and the shift to an insectivorous diet could therefore represent a paedomorphic signal in mammaliaforms adopting a configuration resembling juvenile gomphodontian (and possibly other) cynodonts. The low zygoma height in cf. *Probainognathus*, which likely represents a juvenile individual, compared to the high zygoma in adult individuals, supports the idea of a similar ontogenetic niche shift in this (and possibly other) species. A more widespread ontogenetic change in zygomatic arch height predating the emergence of mammaliaforms could have been a prerequisite for paedomorphic changes in their evolutionary history. Paedomorphosis and, more generally, heterochrony are common mechanisms for skeletal modifications in cynodont and mammaliaform evolution: homoplasies in the mammalian middle ears of Mesozoic mammals[11], the re-acquisition of interpterygoid vacuities in derived non-mammaliaform cynodonts[50], and tooth replacement characteristics in the mammaliaform *Vilevolodon*[51] have been explained as results of heterochronic processes. It is, therefore, possible that the evolution of the zygomatic arch followed a similar trend, although this requires further quantitative testing.

Compared to a carnivorous and herbivorous diet, the adaptation to insectivory could have relaxed the requirements for a strong cranial structure and powerful jaw muscles necessary for feeding, although different insectivorous prey might have required further cranial adaptations[39]. A reduction in size or even loss of the zygomatic arch in modern insectivorous mammals demonstrates that this structure is not necessarily essential or relevant for insectivorous feeding[43].

It is not possible to distinguish whether the size reduction of the zygomatic arch triggered a dietary change towards insectivory or whether dietary adaptation allowed the reduction of the zygoma across the cynodont–mammaliaform transition. Morphofunctional evolution and dietary adaptation likely closely intersected, and these modifications emerged along a continuum of feeding ecomorphology and were closely correlated with other changes in the skull. As shown by previous biomechanical analysis of the mandible, the reduction in size allowed a modification of the jaw joint and middle ear complex[18]. Such miniaturisation

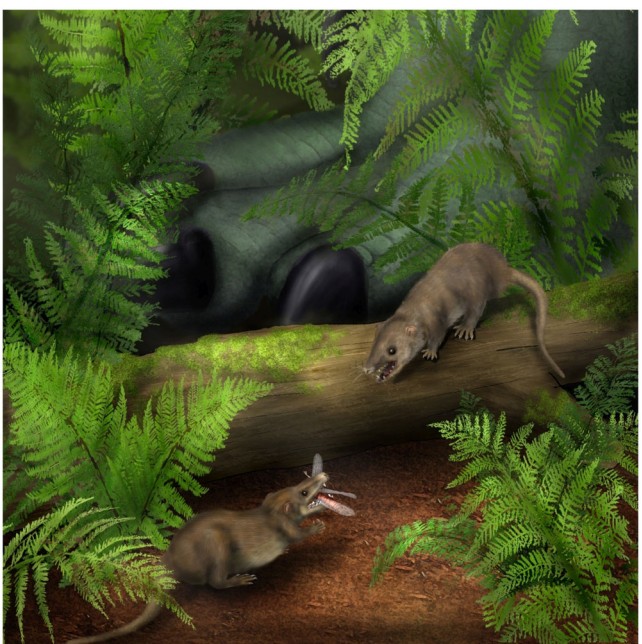

**Fig. 7 Artistic reconstruction of the environmental setting and lifestyle of early mammaliaforms.** Two individuals of *Hadrocodium wui* are shown hunting insect prey illustrating how the adoption of an insectivorous diet and miniaturisation likely played a pivotal role in the functional reorganisation of the cranial skeleton (Image credit: Stephan Lautenschlager).

inevitably restricted the food sources by size and forced the dietary specialisation toward insectivory along the carnivory–insectivory continuum. While the reduction in absolute bite force is a factor of reduced body size, the decrease in mechanical advantage is a likely consequence of insectivory. Low mechanical advantage correlates with faster jaw closing speed, a trait highly relevant to an insectivorous animal. Miniaturisation reduced mechanical advantage, and selective reorganisation of the cranial skeleton facilitated a new dietary ecology (i.e., insectivory) while at the same time relaxing constraints on skull morphology and function, thereby enabling ecological diversification in the rise of mammaliaforms[52].

In summary, the results of the biomechanical analyses do not support a general trend for the increase of cranial strength across the cynodont–mammalian transition but instead provide evidence for a selective functional reorganisation of the cranial skeleton. In particular, the zygoma shows a consistent trend in size reduction and an attendant increase in stress susceptibility. While this likely had a negative initial impact on the structural integrity of the zygoma, the evolution of temporal fasciae potentially provided a bracing system to counter muscle forces and maintained functional performance while reducing material. Stress deflected from the posterior region of the skull in this fashion reduced the loading of the braincase, which in turn could have facilitated the expansion of the braincase in conjunction with brain encephalization along the mammaliaform evolution.

The biomechanical analyses further demonstrate that the efficiency in transfer from muscle forces to bite forces (=mechanical advantage) did not increase across the cynodont–mammaliaform transition. Rather, the low mechanical advantage is consistent with an adaptation of mammaliaform taxa to an insectivorous diet (Fig. 7). Our results suggest that selective functional reorganisation of the cranial skeleton and its companion changes in muscle function, coupled with overall miniaturisation in body size provided a biomechanical environment closely linked with

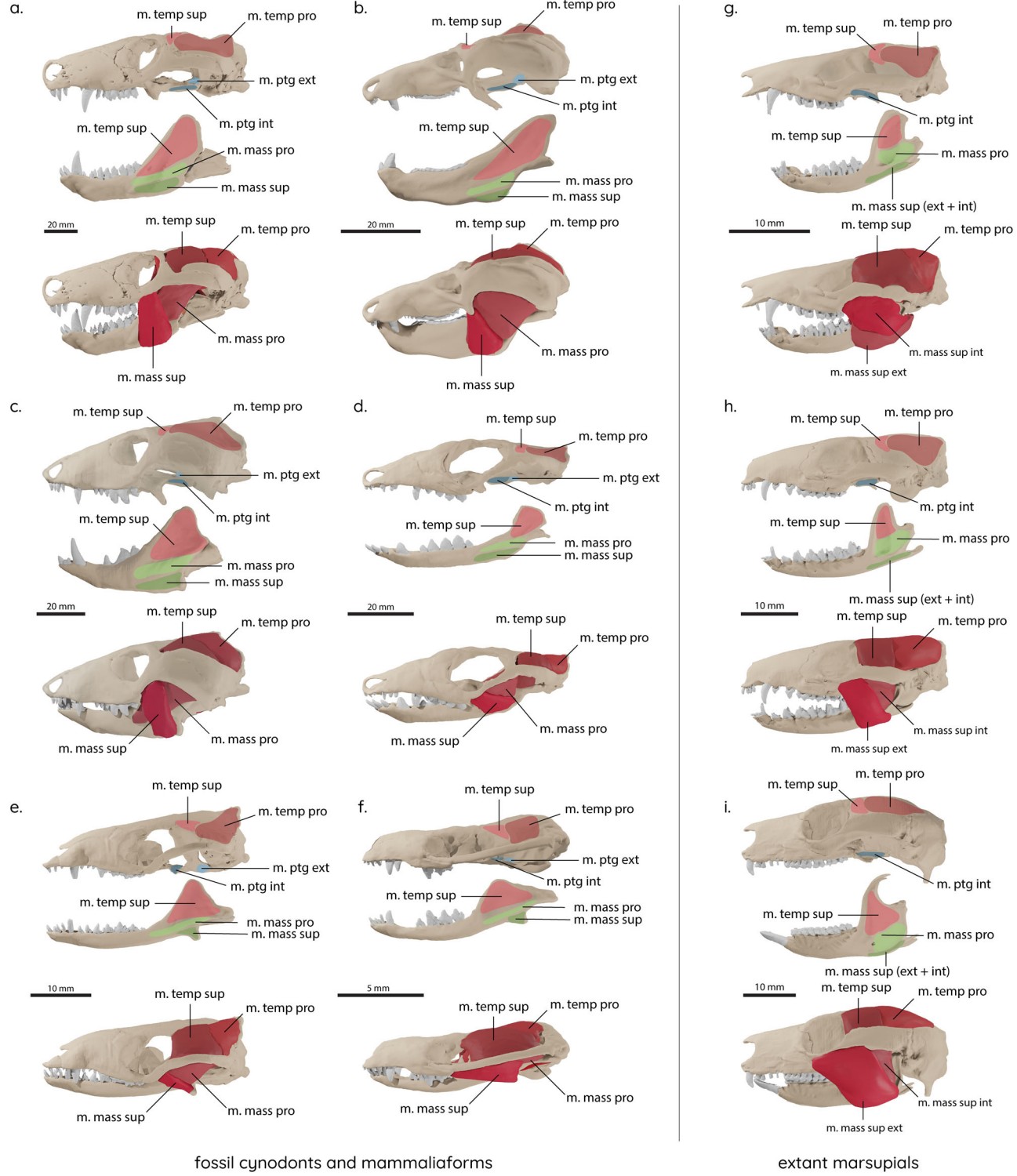

**Fig. 8 Digitally restored cynodont and mammaliaform species.** Digitally restored skull and lower jaw model (top) and digitally reconstructed jaw muscles (bottom). **a** *Thrinaxodon liorhinus*, **b** *Diademodon tetragonus*, **c** *Chiniquodon sanjuanensis*, **d** cf. *Probainognathus* sp., **e** *Morganucodon oehleri*, **f** *Hadrocodium wui*, **g** *Monodelphis domestica*, **h** *Dasyurus hallucatus*, **i** *Petropseudes dahli*. m. mass. pro., m. masseter pars profunda; m. mass. sup., m. masseter pars superficialis; m. ptg. ext., m. pterygoideus externus; m. ptg. int., m. pterygoideus internus; m. temp. pro., m. temporalis pars profunda; m. temp. sup., m. temporalis pars superficialis.

dietary specialisation for insectivory. This setting permitted further morphofunctional modifications, such as the emergence of a novel jaw joint and the subsequent evolution of the detached mammalian middle ear, leading to the successive morphological and ecological diversification of the mammalian lineage.

## Materials and methods

**Specimens and digital models.** Digital models of six fossil cynodonts and mammaliaforms (Fig. 8) were created for this study: *Thrinaxodon liorhinus* (NHMUK PV R 511, 511a, Natural History Museum, London, UK); *Diademodon tetragonus* (BSP 1934 VIII 17/2, Bayerische Staatssammlung für Historische Geologie und Paläontologie, Munich, Germany); *Chiniquodon sanjuanensis* (PVSJ 411,

## Table 1 Calculated jaw adductor muscle forces.

| m. temp. sup | m. temp. pro | m. mass. sup. | m. mass. pro. | m.pter. ext. | m. pter. int. | SUM |
|---|---|---|---|---|---|---|
| *Thrinaxodon liorhinus* | | | | | | |
| 26.4 | 64.0 | 14.4 | 9.5 | 0.5 | 0.12 | 114.92 |
| *Diademodon tetragonus* | | | | | | |
| 360 | 510 | 75 | 428 | 38 | 118 | 1529 |
| *Chiniquodon sanjuanensis* | | | | | | |
| 32.4 | 40.8 | 12.5 | 20.3 | 6.2 | 3.4 | 115.6 |
| cf. *Probainognathus* sp. | | | | | | |
| 4.8 | 9.0 | 4.2 | 5.1 | 1.3 | 3.9 | 28.3 |
| *Morganucodon oehleri* | | | | | | |
| 2.4 | 1.0 | 2.4 | 1.5 | 0.2 | 0.3 | 7.8 |
| *Hadrocodium wui* | | | | | | |
| 0.64 | 0.12 | 0.88 | 0.45 | 0.22 | 0.24 | 2.55 |
| *Monodelphis domestica* | | | | | | |
| 7.2 | 2.0 | 8.5 | 7.4 | 1.1 | 3.3 | 29.5 |
| *Dasyurus hallucatus* | | | | | | |
| 7.6 | 11.2 | 4.6 | 4.4 | 0.6 | 3 | 31.4 |
| *Petropseudes dahli* | | | | | | |
| 20.4 | 33 | 40.8 | 26.6 | 3 | 6.4 | 130.2 |

Muscle loads for each muscle group as used in the biomechanical analyses. All magnitudes in newtons (N).
*m. mass. pro.* m. masseter pars profunda, *m. mass. sup.* m. masseter pars superficialis, *m. ptg. ext.* m. pterygoideus externus, *m. ptg. int.* m. pterygoideus internus, *m. temp. pro.* m. temporalis pars profunda, *m. temp. sup.* m. temporalis pars superficialis.

Museo de Ciencias Naturales, Universidad Nacional de San Juan, Argentina); cf. *Probainognathus* sp. (PVSJ 410); *Morganucodon oehleri* (FMNH CUP 2320, Field Museum of Natural History, Chicago, USA; IVPP 8685, Institute for Vertebrate Palaeontology and Palaeoanthropology) (supplemented by elements from *Morganucodon watsoni* (NHMUK PV M 26144, articulated squamosal and petrosal; NHMUK PV M 92838 & M 92843, isolated quadrates; NHMUK PV M 27410, isolated fragmentary jugal)); *Hadrocodium wui* (IVPP 8275). In addition, models of three extant marsupials were created to provide comparisons for taxa with known diets: *Monodelphis domestica* (Grey short-tailed opossum, Z.2013.185.1, National Museum of Scotland, Edinburgh), an insectivore;[44] *Dasyurus hallucatus* (Northern quoll, TMM M-6921, Texas Memorial Museum, http://digimorph.org/specimens/Dasyurus_hallucatus), a carnivore;[44] *Petropseudes dahli* (Rock ringtail possum, AMNH 183391, American Museum of Natural History, http://digimorph.org/specimens/Petropseudes_dahli), a herbivore[53].

Datasets derived from CT scanning for all specimens were imported into Avizo (version 8, VSG, Visualisation Science Group) for segmentation and removal of taphonomic artefacts. The scanning parameters and a detailed protocol of the restoration process are provided in Lautenschlager et al.[12] as well as in the Supplementary Information (Supplementary Note 1).

To obtain input parameters for the FEA), the jaw adductor musculature was reconstructed digitally[54,55] (Fig. 8). Reconstructions were performed based on osteological correlates indicating muscle attachments and topological criteria. Corresponding attachments of each muscle were linked by simplified point-to-point connections to reconstruct the general muscle arrangement and to identify possible intersections. Subsequently, muscle dimensions and volumes were three-dimensionally modelled according to spatial constraints within the skull. Data obtained from contrast-enhanced CT scanning of *Monodelphis domestica* was consulted to further inform the fossil muscle reconstructions. Full details and discussion of the reconstructed jaw adductor complex for all taxa can be found in Lautenschlager et al.[12]. The final muscle reconstructions were used to calculate muscle forces based on physiological cross-section area[56], which was estimated by dividing the volume of each muscle by its total length.

**Finite element analysis.** For the FEA, all models were imported into Hypermesh (version 11, Altair Engineering) for the creation of solid mesh FE models and the setting of boundary conditions. The skull models were subdivided into six anatomical regions to test for functional independence, largely following the differentiation found by Ackermann and Cheverud[24] and Goswami[57] for modern therians: (1) the narial region (consisting of the premaxillae, maxillae and nasals), (2) the frontal region (prefrontals, frontals, postorbitals, lacrimals), (3) the skull roof (parietal, postparietals, tabulars, supraoccipital), (4) the zygoma (jugals, squamosals, if present quadrates and quadratojugals), (5) the palatal region (pterygoids, palatines, vomer), and (6) the braincase (alisphenoids, orbitosphenoids [if ossified], prootics, exoccipitals, basioccipital, basisphenoid). Although it cannot be ruled out that the crania of cynodonts and mammaliaforms correspond to a different (i.e., 'less mammalian') pattern of modularity and cranial integration, it is important not to confuse morphological modularity as defined by similarity/dissimilarity of skull shape with functional units as tested here. While it would be of interest to compare the morphological and functional modularity of the skulls of a broader range of premammalian cynodonts with divergent dietary specialisations

such as herbivores vs carnivores, this is beyond the scope of this study. Nonetheless, our results provide a baseline for such comparative analyses in the future.

The complete skull models consisted of ca. 2,500,000 tetrahedral elements each. Material properties for bone and teeth were assigned in Hypermesh based on nanoindentation results for small mammal (hedgehog) cranial elements (bone: $E = 12$ GPa, $\upsilon = 0.30$, tooth: $E = 25.0$ GPa, $\upsilon = 0.3$) with all materials treated as isotropic and homogenous[18]. Models were constrained at the quadrate or the squamosal for taxa in which the former had been lost from the cranial structure (eight nodes aligned mediolateral on each side, in x-, y-, and z-direction on the working side and x-, z-direction on the balancing side to allow mediolateral movement). To simulate unilateral biting at different analogous positions, additional dorsoventral constraints (one node each) were applied to the canine and the posteriormost tooth. Further analyses were performed for bilateral biting with constraints applied symmetrically on both canine teeth. Muscle forces were assigned according to the calculations taken from the three-dimensional reconstructions (Table 1). The models were subsequently imported into Abaqus 6.10 (Simulia) for analysis and post-processing. Biomechanical performance for each taxon and each functional module was assessed via contour plot outputs, reaction forces (=bite forces) at the bite points and average stress, strain and displacement values per element.

Biomechanical performance was quantified using FEA contour plots (Figs. 2 and 3), stress distributions for individual cranial regions illustrated by ridgeline plots (Fig. 4), model deformation (Fig. 5), and reaction forces (=bite forces) (Fig. 6) obtained from the FE models. For the quantification of deformation, undeformed and deformed FE models were exported from Abaqus and landmarked in Avizo (Thermo Fisher Scientific, v.8) (see Supplementary Fig. 3). The landmark data was then subjected to a Procrustes and principal component analysis in PAST[58]. Euclidean distances were calculated to quantify the differences between each model pair (undeformed/deformed) for PC1–PC3.

**Reporting summary.** Further information on research design is available in the Nature Portfolio Reporting Summary linked to this article.

## Data availability

All relevant data[59] (three-dimensional osteological and FEA models, landmark data) are available at the University of Bristol data repository, data.bris, at https://doi.org/10.5523/bris.21ypbecdc308m2c32nqpown5y0.

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

## Acknowledgements

We thank P. Brewer, S. Chapman (NHM, London), O. Rauhut, G. Rössner (BSPG, Munich), K. Angielczyk and W. Simpson (FMNH, Chicago), G. Hantke and A. Kitchener (NMS, Edinburgh) for access to specimens in their care. T. Rowe and J. Maisano (University of Texas, Austin) generously provided digital datasets of specimens. A.I. Neander (University of Chicago), G. Rössner (BSPG, Munich), D. Sykes (NHM, London), K. Robson Brown (University of Bristol), O. Katsamenis and M. Mavrogordato (University of Southampton) assisted with CT scanning. J.A. Hopson (University of Chicago) is thanked for the discussion. This work was funded by NERC grants NE/K01496X/1 (to E.J.R.) and NE/K013831/1 (to M.J.F.).

## Author contributions

S.L., M.J.F., Z.-X.L., P.G., and E.J.R. conceived and designed the study. S.L., Z.-X.L., P.G., and E.J.R. arranged the logistics of specimens for CT scanning and collected CT data. Z.-X.L. provided access to additional specimens and data. S.L. processed CT data, performed digital restorations and reconstructions, and performed computational analyses. C.M.B. collected landmark data and performed GMM analyses. S.L., M.J.F., Z.-X.L., C.M.B., P.G., and E.J.R. equally contributed to the analysis of results. S.L. prepared the main text, figures, and supplementary data. S.L., M.J.F., Z.-X.L., C.M.B., P.G., and E.J.R. equally contributed to editing, commenting, and revising the manuscript and figures. M.J.F. and E.J.R. acquired funding.

## Competing interests

The authors declare no competing interests.
