## [Peer Review File · Communications Biology]

Reviewers' comments:

Reviewer #1 (Remarks to the Author):

This manuscript is an important step towards moving our understanding of mammalian origins away from the hand-waving of 20th century accounts, and actually testing hypotheses with some semblance of analytical rigor. "Simplification of the cynodont skull was to strengthen biting" is one of those statements that has appeared in some form or another in countless papers, based largely on gut feelings of what should make sense instead of hard data. The fact that this neat little story doesn't match biological reality is not something I find hugely surprising, but it is critical that it be demonstrated quantitatively, and the authors do an admirable job with an excellent sample of the relevant taxa. In general, I believe this manuscript is well-deserving of publication in the journal and hopefully will promote a new look at various other "old chestnuts" of mammal evolution.

My criticisms of the manuscript are mostly fairly minor, see line edits below. I do take issue with the statements on paedomorphosis, which seem like tossed-in asides rather than serious proposals. As it so happens, I believe the authors are exactly correct and that heterochrony underlies many changes in the cynodont skull during their miniaturization phase, but this idea is presented devoid of context (or even any real referencing, and there is a healthy body of literature that could be cited as support, e.g. Barghusen & Hopson, 1970). Further explication or at least some citations of where this idea is coming from are needed.

I can find no basis for the dietary categories of the sampled taxa either in the manuscript or supplementary materials; this must be specified.

Line 46: "generate" should be "generated"

Figure 1: There is no evidence that *Probelesodon* crownward of *Chiniquodon*, the two are very similar and both fall in the clade *Chiniquodontidae*, supported by such synapomorphies as a large extranasal process of the premaxilla and pterygoid elongation (indeed, *Probelesodon* has been treated as a junior synonym of *Chiniquodon* in almost all cynodont taxonomic literature of the past 20 years, although I believe there is good reason to believe that *lewisii* is at least specifically distinct from *theotonicus*. There is no phylogenetic data supporting *sanjuanensis* forming a monophyletic *Probelesodon* with *lewisii* to the exclusion of *Chiniquodon theotonicus*, however).

Line 86: "*Hadrocodium*" should be italicized

Line 93: "P." should be italicized

Line 154: *Pachygenelus* is an odd choice of exemplar here, as this taxon is not included in your study sample (and given substantial uncertainty as to details of tritheledontid taxonomy, it is uncertain what you are actually talking about when you say "*Pachygenelus*").

Line 190: This is almost certainly the case.

Line 206: "observed" should be "observed in"

Figure 8: I think I have complained about this in reviews of earlier uses of this model, but the anterior snout region of your *Probelesodon sanjuanensis* reconstruction remains extremely far from reality, with an unbelievably robust premaxilla and bizarrely retracted naris. You really do need to fix this if it's going to be used in every FEA-based study of mammal origins.

Reviewer #2 (Remarks to the Author):

General comments: This manuscript uses finite element analyses to investigate how stress distributions in the skull during biting change during the cynodont-to-mammal evolutionary transition. It's an interesting problem to consider because there's a lot things happening during

this interval of mammalian history, including simplification of the skull through the loss or combination of some elements, reduction in body size, potential dietary changes, the elaboration of a mammalian jaw muscle system, and changes related to the incorporation of element of the jaw into the middle ear. The authors find that skull strength doesn't change all that much across the transition, refuting previous hypotheses that there was an increase in skull strength, and that some areas of the skull, such as the zygomatic arch actually become weaker in at least some of the taxa they sampled. There's also changes in distribution of stress in the braincase that may be related to increases in brain size that take place during this transition. Overall, the paper is well written, and has a lot of good content. However, there's also a few issues that exist, although given the nature of the work, I'm not sure how easy they will be to address.

1) The taxon sample is low considering some of the generalizations that are being made. I'll follow that comment up by saying that I know that it not a trivial task to find specimens that are suitable for this kind of work, or to make the models that go into the analyses, so I wouldn't expect the study to include dozens of specimens. Likewise, the included taxa do hit some important points in the cynodont-mammaliaform transition. However, the fact that there's just one herbivorous taxon in the dataset really limits what they can say about the importance of diet in the transition. There's also a hint that more basal cynodont herbivores might have been doing something different than mammalian herbivores when it comes to bite force and mechanical advantage, but I don't think there's a way to say this with any certainty with only one cynodont herbivore and one mammal herbivore in the dataset.

2) Given that some key parts of the skull of *Hadrocodium* had to be reconstructed, particularly the zygomatic arch, I would be interested to see more of an effort to determine whether the odd analytical results for that taxon stem (at least in part) from the reconstruction process. One way to do this might be to try a sensitivity analysis utilizing a couple different alternative zygomatic arch morphologies to see if they have much impact on the results.

3) A somewhat minor point, but *Probelesodon* is generally considered to be a junior synonym of *Chiniquodon* (see Abdala and Giannini 2002), so if the authors are going to use the name *Probelesodon*, I think they need to include some justification for why they think it is distinct from *Chiniquodon*.

4) The *Monodelphis* specimen should be accessioned.

Line 35: I recommend revising this slightly to note that there is evidence of simplification earlier in synapsid history than in cynodonts, although the trend for simplification is stronger in cynodonts.

Line 46: change to generated

Fig. 1: Is there a reason why you didn't include measurements for the moderns species on the tree? It seems like they would be useful for comparison to the fossil taxa. Also, the Eutheria label is kind of confusing because it could be interpreted as indicating that the modern species are eutherians. I think it would be fine to leave that off. Finally, usually *Probelesodon* is treated as a synonym of *Chiniquodon*, and this seems to be an underlying assumption for the Pacheo et al. phylogeny you use as the main cynodont topology (i.e., they don't include *Probelesodon* as an OTU). So what is the rationale for treating it as a distinct taxon here?

Line 91: replace increased with higher

Fig. 6: I understand the attraction of having carnivores as the red hull and herbivores as green, but I worry about whether the color scheme will be accessible to people with colorblindness? Have you tested the specific color scheme used in this figure to see if the colors are still distinguishable?

Line 113: The generalization you make about herbivorous taxa doesn't really seem to hold because the one herbivorous cynodont that you include, *Diademodon*, has a low mechanical advantage (although it does seem to have high bite force on account of being large). I think this is a significant point because its jaw has undergone less extensive transformation than some of the later members of *Probainognathia* and mammaliaforms in particular, so maybe this is indicative of

a difference in how more basal cynodonts approached herbivory compared to the herbivorous mammals you include. Ultimately it would be nice if you could include some additional herbivorous cynodonts in the analysis to try to better characterize what's going on, but I also realize that it is a fairly significant undertaking to get the kind of data needed for the analyses.

Line 170: Do you think the extreme small size of *Hadrocodium* might be a factor here as well? Specifically, because it was very small, and its bite force was absolutely very low, could it get away with relatively high stresses in its skull because they were still very small in absolute terms?

Line 183: Something else that might be worth considering here is how the changes in the masseter musculature might have impacted jaw function beyond force generation. Could those changes have resulted in new means of control of the jaw that could have improved feeding efficiency without having to increase bite force? In other words, instead of using a relatively simple set of jaw muscles and jaw movements to brute force the processing of food, could the changing musculature have facilitated new lines of jaw function that could accomplish similar or better degrees of food processing with less force by allowing the available force to be applied in a more effective way?

Line 221: remove were

Line 276: Can the *Monodelphis* specimen be cataloged? It seems like it would be appropriate to do so because of its role in this paper.

Line 351: William Simpson should also be thanked for access to the FMNH *Morganucodon* specimen.

377: change date to data

Supplement: restoration of specimens: Maybe include a few standard photos of the specimens you used to show show their original state. I think that will give readers a sense of the degree to which restoration was necessary in the different cases.

Supplement, *Diademodon* restoration: More detail on the reconstruction of the tooth crowns would be useful here. In particular, did you start with a model of another *Diademodon* tooth? If the teeth were completely fabricated, how closely do the modeled teeth match the morphology of real *Diademodon* teeth?

Supplement, *Probelesodon*: As noted above, *Probelesodon* is generally considered to be a synonym of *Chiniquodon*. The Abdala and Giannini (2002) paper you cite in this section is the source of that synonymy, and they include the specimen you're using as part of their analysis. I recommend using *Chiniquodon* as the genus name instead. If you continue to use *Probelesodon*, I think you should justify that taxonomic decision, presumably as a separate section of the supplement.

Supplement, *Probainognathus*: The potential juvenile status of this specimen is interesting. The adult skull morphology of *Probainognathus* is superficially pretty similar to *Chiniquodon*, so the somewhat more mammal like morphology of the smaller specimen raises questions about how much advanced cynodonts/early mammaliaforms were changing size, morphology, and potentially function along a conserved allometric pattern. You note the somewhat similar change in skull proportions reported for *Exaeretodon* by Wynd et al. In the main text, so I could see it being worthwhile to note that there is something potentially similar going on with *Probainognathus* there.

Supplement, *Hadrocodium*: The fact that you had to reconstruct the zygomatic arch here, without as much of a guide as was available for *Morganucodon* (i.e., using elements preserved in *M. watsoni*) as a guide, how much do you think alternative morphologies of the zygomatic arch might impact the unusual results for *Hadrocodium*? In other words, how different would the zygomatic arch have to be in order to have the specimen behave more 'normally' in the FEA analyses? Or is that different behavior the result of other aspects of the model that are unrelated or only peripherally related to the zygomatic arch morphology? At minimum, I think it would be good to note in the main text that some parts of the skull that seem to play an important role in your

narrative had to be reconstructed for Hadrocodium, and this might in part explain why its results stand out from the others so much. Ideally, I would like it if you could do a sensitivity analysis looking at the effects of alternative zygomatic arch morphologies in Hadrocodium to see how much they impact the results, and/or include more information about what aspects of the model seem to be driving the unusual results (which in turn might show that the zygomatic arch morphology is unlikely to have a major effect).

Supplement, Monodelphis: This is just repeating one of my earlier comments, but I think it would be appropriate to accession the Monodelphis specimen given that it is being used formally in a publication.

Response to Reviewers' comments:

Reviewer #1 (Remarks to the Author):

- This manuscript is an important step towards moving our understanding of mammalian origins away from the hand-waving of 20th century accounts, and actually testing hypotheses with some semblance of analytical rigor. “Simplification of the cynodont skull was to strengthen biting” is one of those statements that has appeared in some form or another in countless papers, based largely on gut feelings of what should make sense instead of hard data. The fact that this neat little story doesn't match biological reality is not something I find hugely surprising, but it is critical that it be demonstrated quantitatively, and the authors do an admirable job with an excellent sample of the relevant taxa. In general, I believe this manuscript is well-deserving of publication in the journal and hopefully will promote a new look at various other “old chestnuts” of mammal evolution.

Lautenschlager et al: We thank the reviewer for such a positive evaluation of our study and are pleased to see the relevance of the topic highlighted.

- My criticisms of the manuscript are mostly fairly minor, see line edits below. I do take issue with the statements on paedomorphosis, which seem like tossed-in asides rather than serious proposals. As it so happens, I believe the authors are exactly correct and that heterochrony underlies many changes in the cynodont skull during their miniaturization phase, but this idea is presented devoid of context (or even any real referencing, and there is a healthy body of literature that could be cited as support, e.g. Barghusen & Hopson, 1970). Further explication or at least some citations of where this idea is coming from are needed.

Lautenschlager et al: We have added a longer discussion of the possible role of heterochronic processes in shaping the mammaliaform/mammalian cranial morphology, including relevant studies (note, heterochrony is not discussed in Barghusen & Hopson, 1970 so not cited here). (lines 185-209):

“The change in zygomatic arch height parallels ontogenetic niche shifts observed in gomphodontian cynodonts. For example, smaller individuals (inferred to be juveniles) of *Exaeretodon argentinus* and *E. riograndensis* had a low zygoma adapted to crushing-dominated feeding whereas larger individuals (inferred to be adults/older ontogenetic stages) showed a high zygoma adapted to a chewing-dominated feeding style⁴⁷. A similar trend has been found tentatively also in the more basal cynodonts *Galesaurus* and *Thrinaxodon*^{48,49}. A specialisation of the musculoskeletal system towards a crushing function is commonly correlated with an adaptation to faunivory (including insectivory). The skeletal miniaturisation, reorganisation of individual skull regions (including the zygoma), and the shift to an insectivorous diet could therefore represent a paedomorphic signal in mammaliaforms adopting a configuration of juvenile gomphodontian (and possibly other) cynodonts. The low zygoma height in cf. *Probainognathus*, which likely represents a juvenile individual, compared to the high zygoma in adult individuals supports the idea of a similar ontogenetic niche shift in this (and possibly other) species. A more widespread ontogenetic change in zygomatic arch height predating the emergence of mammaliaforms could have been a prerequisite for paedomorphic changes in their evolutionary history.

Paedomorphosis and, more generally, heterochrony are common mechanisms for skeletal modifications in cynodont and mammaliaform evolution: homoplasies in the mammalian middle ears of Mesozoic mammals¹¹, the re-acquisition of interpterygoid vacuities in derived non-mammaliaform cynodonts⁵⁰, and tooth replacement characteristics in the mammaliaform *Vilevolodon*⁵¹ have been explained as results of heterochronic processes. It is therefore possible that the evolution of the zygomatic arch followed a similar trend, although this requires further quantitative testing. “

- I can find no basis for the dietary categories of the sampled taxa either in the manuscript or supplementary materials; this must be specified.

Lautenschlager et al: Reference information on the dietary categories is now included in the caption for figure 1.

- Line 46: “generate” should be “generated”

Lautenschlager et al: Corrected

- Figure 1: There is no evidence that *Probelesodon* crownward of *Chiniquodon*, the two are very similar and both fall in the clade *Chiniquodontidae*, supported by such synapomorphies as a large extranasal process of the premaxilla and pterygoid elongation (indeed, *Probelesodon* has been treated as a junior synonym of *Chiniquodon* in almost all cynodont taxonomic literature of the past 20 years, although I believe there is good reason to believe that *lewisii* is at least specifically distinct from *theotonicus*. There is no phylogenetic data supporting *sanjuanensis* forming a monophyletic *Probelesodon* with *lewisii* to the exclusion of *Chiniquodon theotonicus*, however).

Lautenschlager et al: *Probelesodon* has been removed in the figure (see also similar suggestion from reviewer 2 below) and the specimen used in the study is now treated as *Chiniquodon* to reflect the up-to-date taxonomy.

- Line 86: “*Hadrocodium*” should be italicized

Lautenschlager et al: Corrected

- Line 93: “P.” should be italicized

Lautenschlager et al: Corrected

- Line 154: *Pachygenelus* is an odd choice of exemplar here, as this taxon is not included in your study sample (and given substantial uncertainty as to details of *tritheledontid* taxonomy, it is uncertain what you are actually talking about when you say “*Pachygenelus*”).

Lautenschlager et al: We have removed the reference to *Pachygenelus* to avoid any confusion.

- Line 190: This is almost certainly the case.

Lautenschlager et al: *Changed to “We further note that the specimen of cf. Probainognathus is very likely a juvenile;”*

- Line 206: “observed” should be “observed in”

Lautenschlager et al: *Corrected*

- Figure 8: I think I have complained about this in reviews of earlier uses of this model, but the anterior snout region of your Probelesodon sanjuanensis reconstruction remains extremely far from reality, with an unbelievably robust premaxilla and bizarrely retracted naris. You really do need to fix this if it’s going to be used in every FEA-based study of mammal origins.

Lautenschlager et al: *Yes, you are absolutely correct. We had adjusted the model following a previous review suggestion already but it seems the old reconstruction snuck into this figure. This has been corrected now.*

Reviewer #2 (Remarks to the Author):

- General comments: This manuscript uses finite element analyses to investigate how stress distributions in the skull during biting change during the cynodont-to-mammal evolutionary transition. It's an interesting problem to consider because there's a lot of things happening during this interval of mammalian history, including simplification of the skull through the loss or combination of some elements, reduction in body size, potential dietary changes, the elaboration of a mammalian jaw muscle system, and changes related to the incorporation of elements of the jaw into the middle ear. The authors find that skull strength doesn't change all that much across the transition, refuting previous hypotheses that there was an increase in skull strength, and that some areas of the skull, such as the zygomatic arch actually become weaker in at least some of the taxa they sampled. There's also changes in distribution of stress in the braincase that may be related to increases in brain size that take place during this transition. Overall, the paper is well written, and has a lot of good content. However, there's also a few issues that exist, although given the nature of the work, I'm not sure how easy they will be to address.

Lautenschlager et al: We thank the reviewer for the positive and constructive feedback.

- The taxon sample is low considering some of the generalizations that are being made. I'll follow that comment up by saying that I know that it's not a trivial task to find specimens that are suitable for this kind of work, or to make the models that go into the analyses, so I wouldn't expect the study to include dozens of specimens. Likewise, the included taxa do hit some important points in the cynodont-mammaliaform transition. However, the fact that there's just one herbivorous taxon in the dataset really limits what they can say about the importance of diet in the transition. There's also a hint that more basal cynodont herbivores might have been doing something different than mammalian herbivores when it comes to bite force and mechanical advantage, but I don't think there's a way to say this with any certainty with only one cynodont herbivore and one mammal herbivore in the dataset.

Lautenschlager et al: We agree that more taxon sampling is always desirable but the realities of accessing and digitising specimens, that need to be preserved well enough to be meaningful are not always allowing this (and we are glad that the reviewer is agreeing with this as well). With regard to dietary diversity, we do actually not make any substantial generalisations regarding herbivorous taxa. The only reference is in line 101 ("Herbivorous taxa, conversely, have the highest absolute and relative bite forces") and we have now tempered this statement with the following sentence ("...but it should be noted that our sample includes only one fossil and one modern herbivorous species.").

- 2) Given that some key parts of the skull of Hadrocodium had to be reconstructed, particularly the zygomatic arch, I would be interested to see more of an effort to determine whether the odd analytical results for that taxon stem (at least in part) from the reconstruction process. One way to do this might be to try a sensitivity analysis utilizing a couple different alternative zygomatic arch morphologies to see if they have much impact on the results.

Lautenschlager et al: *Following the unusual results for Hadrocodium we had indeed performed sensitivity tests using differently reconstructed zygomatic arch morphologies. However, regardless of the reconstructed shape, stress values were always recovered as substantially higher than in any of the other models. A figure showing the different reconstructions and quantification is now included in the supplementary material and a brief explanation has been added in the main text (lines 90-94):*

*“Additional tests were performed for *H. wui*, due to the high stress magnitudes experienced in the zygomatic region and the fact that the zygoma had to be reconstructed to a large extent for this specimen. However, regardless of the morphology of the reconstructed zygomatic region recorded stress values were more than 100% higher than in the other species (Fig. S2).”*

- 3) A somewhat minor point, but *Probelesodon* is generally considered to be a junior synonym of *Chiniquodon* (see Abdala and Giannini 2002), so if the authors are going to use the name *Probelesodon*, I think they need to include some justification for why they think it is distinct from *Chiniquodon*.

Lautenschlager et al: *The specimen included in the study is now referred to as *Chiniquodon* throughout the text to reflect the up-to-date taxonomy.*

- 4) The *Monodelphis* specimen should be accessioned.

Lautenschlager et al: *Confirming with the museum, the specimen has been accessioned (Z.2013.185.1).*

- Line 35: I recommend revising this slightly to note that there is evidence of simplification earlier in synapsid history than in cynodonts, although the trend for simplification is stronger in cynodonts.

Lautenschlager et al: *We rephrased the sentence to make it more inclusive regarding synapsids in general (lines 33-35):*

“These simplification events can be found across different tetrapod lineages⁵⁻⁷ and throughout synapsid history, including mammals and their synapsid, pre-mammaliaform cynodont precursors (referred to as cynodonts hereafter)⁸.”

- Line 46: change to generated

Lautenschlager et al: *Corrected.*

- Fig. 1: Is there a reason why you didn't include measurements for the moderns species on the tree? It seems like they would be useful for comparison to the fossil taxa. Also, the Eutheria label is kind of confusing because it could be interpreted as indicating that the modern species are eutherians. I think it would be fine to leave that off. Finally, usually *Probelesodon* is treated as a synonym of *Chiniquodon*, and this seems to be an underlying assumption for the Pacheo et al. phylogeny you use as the main cynodont topology (i.e.,

they don't include Probelesodon as an OTU). So what is the rationale for treating it as a distinct taxon here?

Lautenschlager et al: Figure has been changed as suggested: (i) Eutheria label has been removed to avoid confusion; (ii) Probelesodon has been removed and specimen included in the study is now referred to as Chiniquodon to reflect the up-to-date taxonomy. (iii) relative zygomatic height and braincase width measurements have been included for the modern taxa.

- Line 91: replace increased with higher

Lautenschlager et al: Changed as suggested.

- .Fig. 6: I understand the attraction of having carnivores as the red hull and herbivores as green, but I worry about whether the color scheme will be accessible to people with colorblindness? Have you tested the specific color scheme used in this figure to see if the colors are still distinguishable?

Lautenschlager et al: An important point and as with the FEA contour plots and other figures, we did use accessible colours (see colour-blind view below).

- Line 113: The generalization you make about herbivorous taxa doesn't really seem to hold because the one herbivorous cynodont that you include, Diademodon, has a low mechanical advantage (although it does seem to have high bite force on account of being large). I think this is a significant point because its jaw has undergone less extensive transformation than some of the later members of Probaiongnathia and mammaliaforms in particular, so maybe this is indicative of a difference in how more basal cynodonts approached herbivory compared to the herbivorous mammals you include. Ultimately it would be nice if you could include some additional herbivorous cynodonts in the analysis to try to better characterize

what's going on, but I also realize that it is a fairly significant undertaking to get the kind of data needed for the analyses.

Lautenschlager et al: *As noted above, we do actually not make any substantial references to the herbivorous taxa and their impact on the results. Our main comparison is between carnivorous and insectivorous species, with the latter category appearing to be the most relevant one to explain the morphofunctional transformations. Where we refer to herbivorous species, we have added a brief qualifying statement that the sample size in this category is indeed low. (lines 101-102)*

"Herbivorous taxa, conversely, have the highest absolute and relative bite forces; but it should be noted that our sample includes only one fossil and one modern herbivorous species."

- Line 170: Do you think the extreme small size of *Hadrocodium* might be a factor here as well? Specifically, because it was very small, and its bite force was absolutely very low, could it get away with relatively high stresses in its skull because they were still very small in absolute terms?

Lautenschlager et al: *An interesting idea. Absolute stresses are actually quite high in *Hadrocodium* (possibly due to the larger jaw muscles) despite the low bite forces so it doesn't seem to be a size effect. A similar pattern would be expected in the somewhat larger *Morganucodon* but is not visible. While size seems to be an evolutionary important factor more generally, we see the relevance rather in the shift to an insectivorous diet here, which requires smaller absolute bite forces.*

- Line 183: Something else that might be worth considering here is how the changes in the masseter musculature might have impacted jaw function beyond force generation. Could those changes have resulted in new means of control of the jaw that could have improved feeding efficiency without having to increase bite force? In other words, instead of using a relatively simple set of jaw muscles and jaw movements to brute force the processing of food, could the changing musculature have facilitated new lines of jaw function that could accomplish similar or better degrees of food processing with less force by allowing the available force to be applied in a more effective way?

Lautenschlager et al: *A very good point. We have now added more discussion on this (lines 160-165):*

"The reorientation of the jaw musculature (in particular the masseter complex) could have had another advantage, unrelated to maximising bite force. Greater mobility of the mandible^{10,13} and precise tooth occlusion⁴⁰ may have required precise muscle control to allow more complex jaw movements, including long-axis rotation (perpendicular to the jaw hinge axis). Such rolling of the jaw was recently identified as an essential mechanism for tribosphenic molar function⁴¹."

- Line 221: remove were

Lautenschlager et al: *Corrected.*

- Line 276: Can the Monodelphis specimen be cataloged? It seems like it would be appropriate to do so because of its role in this paper.

Lautenschlager et al: *Confirming with the museum, the specimen has been accessioned (Z.2013.185.1).*

- Line 351: William Simpson should also be thanked for access to the FMNH Morganucodon specimen.

Lautenschlager et al: *Added to the acknowledgements.*

- 377: change date to data

Lautenschlager et al: *Corrected.*

- Supplement, Diademodon restoration: More detail on the reconstruction of the tooth crowns would be useful here. In particular, did you start with a model of another Diademodon tooth? If the teeth were completely fabricated, how closely do the modeled teeth match the morphology of real Diademodon teeth?

Lautenschlager et al: *We have added more information to the respective section in the supplement: "The tooth crowns of both canine teeth in the skull were not preserved in the specimen and were reconstructed using the mandibular canine teeth and the size and position of the mandible and mandibular dentition as guidance."*

- Supplement, Probelesodon: As noted above, Probelesodon is generally considered to be a synonym of Chiniquodon. The Abdala and Giannini (2002) paper you cite in this section is the source of that synonymy, and they include the specimen you're using as part of their analysis. I recommend using Chiniquodon as the genus name instead. If you continue to use Probelesodon, I think you should justify that taxonomic decision, presumably as a separate section of the supplement.

Lautenschlager et al: *The specimen included in the study is now referred to as Chiniquodon throughout the text to reflect the up-to-date taxonomy.*

- Supplement, Probainognathus: The potential juvenile status of this specimen is interesting. The adult skull morphology of Probainognathus is superficially pretty similar to Chiniquodon, so the somewhat more mammal like morphology of the smaller specimen raises questions about how much advanced cynodonts/early mammaliaforms were changing size, morphology, and potentially function along a conserved allometric pattern. You note the somewhat similar change in skull proportions reported for Exaeretodon by Wynd et al. In the main text, so I could see it being worthwhile to note that there is something potentially similar going on with Probainognathus there.

Lautenschlager et al: *Following a similar suggestion from another reviewer (see above) we have added a longer discussion of the possible role of heterochronic processes in shaping the*

mammaliaform/mammalian cranial morphology and a possible ontogenetic niche shift in Probainognathus. (lines 185-206):

“The change in zygomatic arch height parallels ontogenetic niche shifts observed in gomphodontian cynodonts. For example, smaller individuals (inferred to be juveniles) of *Exaeretodon argentinus* and *E. riograndensis* had a low zygoma adapted to crushing-dominated feeding whereas larger individuals (inferred to be adults/older ontogenetic stages) showed a high zygoma adapted to a chewing-dominated feeding style⁴⁷. A similar trend has been found tentatively also in the more basal cynodonts *Galesaurus* and *Thrinaxodon*^{48,49}. A specialisation of the musculoskeletal system towards a crushing function is commonly correlated with an adaptation to faunivory (including insectivory). The skeletal miniaturisation, reorganisation of individual skull regions (including the zygoma), and the shift to an insectivorous diet could therefore represent a paedomorphic signal in mammaliaforms adopting a configuration of juvenile gomphodontian (and possibly other) cynodonts. The low zygoma height in cf. *Probainognathus*, which likely represents a juvenile individual, compared to the high zygoma in adult individuals supports the idea of a similar ontogenetic niche shift in this (and possibly other) species. A more widespread ontogenetic change in zygomatic arch height predating the emergence of mammaliaforms could have been a prerequisite for paedomorphic changes in their evolutionary history. Pedomorphosis and, more generally, heterochrony are common mechanisms for skeletal modifications in cynodont and mammaliaform evolution: homoplasies in the mammalian middle ears of Mesozoic mammals¹¹, the re-acquisition of interpterygoid vacuities in derived non-mammaliaform cynodonts⁵⁰, and tooth replacement characteristics in the mammaliaform *Vilevolodon*⁵¹ have been explained as results of heterochronic processes. It is therefore possible that the evolution of the zygomatic arch followed a similar trend, although this requires further quantitative testing. “

- Supplement, Hadrocodium: The fact that you had to reconstruct the zygomatic arch here, without as much of a guide as was available for Morganucodon (i.e., using elements preserved in *M. watsoni*) as a guide, how much do you think alternative morphologies of the zygomatic arch might impact the unusual results for Hadrocodium? In other words, how different would the zygomatic arch have to be in order to have the specimen behave more ‘normally’ in the FEA analyses? Or is that different behavior the result of other aspects of the model that are unrelated or only peripherally related to the zygomatic arch morphology? At minimum, I think it would be good to note in the main text that some parts of the skull that seem to play an important role in your narrative had to be reconstructed for Hadrocodium, and this might in part explain why its results stand out from the others so much. Ideally, I would like it if you could do a sensitivity analysis looking at the effects of alternative zygomatic arch morphologies in Hadrocodium to see how much they impact the results, and/or include more information about what aspects of the model seem to be driving the unusual results (which in turn might show that the zygomatic arch morphology is unlikely to have a major effect).

Lautenschlager et al: *As noted above, we had performed sensitivity tests using differently reconstructed zygomatic arch morphologies. However, regardless of the reconstructed shape, stress values were always recovered as substantially higher than in any of the other models. A figure showing the different reconstructions and quantification is now included in the supplementary material and a brief explanation has been added in the main text (lines 90-94):*

“Additional tests were performed for *H. wui*, due to the high stress magnitudes experienced in the zygomatic region and the fact that the zygoma had to be reconstructed to a large extent for this specimen. However, regardless of the morphology of the reconstructed zygomatic region recorded stress values were more than 100% higher than in the other species (Fig. S2).”

- Supplement, Monodelphis: This is just repeating one of my earlier comments, but I think it would be appropriate to accession the Monodelphis specimen given that it is being used formally in a publication.

Lautenschlager et al: *Confirming with the museum, the specimen has been accessioned (Z.2013.185.1).*

REVIEWERS' COMMENTS:

Reviewer #2 (Remarks to the Author):

I thank the authors for their revisions to the manuscript, and their detailed responses to the reviews. In my opinion, they have done a good job of addressing the comments provided by me and the other reviewer, and I only have two very minor suggestions for the current version of the manuscript.

1) At line 194, I recommend rewording things a little so that it doesn't sound like you're suggesting the mammaliaforms are using a heterochronic pattern inherited from gomphodonts given that gomphodonts are a separate branch of cynodont phylogeny. Changing to something like "adopting a configuration resembling juvenile gomphodontian..." would probably be sufficient. Alternatively, you could more strongly underscore the possibility that this is a deeper pattern present in cynodonts that both cynognathians (including gomphodonts) and probainognathians (including mammaliaforms) inherited from an older cynodont ancestor.

2) Line 85 should be changed to *C. sanjuanensis*